# The Policy-gradient Placement and Generative Routing Neural Networks for Chip Design

**Ruoyu Cheng[1], Xianglong Lyu[1], Yang Li[1], Junjie Ye[2], Jianye Hao[2], Junchi Yan[1*]**
[1]Department of Computer Science and Engineering, Shanghai Jiao Tong University
[2]Huawei Noah's Ark Lab
{roy_account,kyle_lyu,emiyali,yanjunchi}@sjtu.edu.cn
{yejunjie4, haojianye}@huawei.com

## Abstract

Placement and routing are two critical yet time-consuming steps of chip design in modern VLSI systems. Distinct from traditional heuristic solvers, this paper on one hand proposes an RL-based model for mixed-size macro placement, which differs from existing learning-based placers that often consider the macro by coarse grid-based mask. While the standard cells are placed via gradient-based GPU acceleration. On the other hand, a one-shot conditional generative routing model, which is composed of a special-designed input-size-adapting generator and a bi-discriminator, is devised to perform one-shot routing to the pins within each net, and the order of nets to route is adaptively learned. Combining these techniques, we develop a flexible neural pipeline, which to our best knowledge, is the first joint placement and routing network without involving any traditional heuristic solver. Experimental results on chip design benchmarks showcase the effectiveness of our approach. Source code will be made publicly available at: https://github.com/Thinklab-SJTU/EDA-AI

## 1 Introduction

The scale of integrated circuits (ICs) has been enlarged dramatically, posing a challenge to the existing Electronic Design Automation (EDA) technologies. The increasing circuit density raises additional issues for VLSI placers and routers as the component size of modern VLSI design continues to drop and on-chip connectivity becomes increasingly sophisticated. Due to growing on-chip connectivity, concentrated needs and restricted resources, modern designs are prone to congestion issues and wirelength minimization, which has become a critical task at every stage of the chip design process. Accordingly, placement and routing that physically arranges the locations and the routes of nets become more crucial in modern VLSI systems, which are also very time-consuming.

The components of a netlist, including macros and standard cells, are mapped to positions on the chip layout by placement, with standard cells being basic logic cells (e.g. logic gates) and macros as functional blocks (e.g. SRAMs). Moreover, the increasingly extensive use of intellectual property (IP) modules and pre-designed macro blocks make mixed-size placement an indispensable part of physical design. The goal of placement is to optimize power, performance, and area (PPA) metrics meanwhile obeying the constraints e.g. placement density and routing congestion. It is worth noting that, in this paper we address the global routing problem, while the detailed routing involves more physical constraints, and is beyond the scope of this paper.

Global routing, on the other hand, creates routing channels inside a layout based on the placement assignment by connecting pins of positioned IC components to a net while adhering to technological

---

*The SJTU authors are also with MoE Key Lab of Artificial Intelligence, SJTU. Junchi Yan is the correspondence author who is also with Shanghai AI Laboratory.

limits. It is tightly coupled with the placement task, as an excellent placement solution can result in better chip area utilization, timing performance, and routability etc. The objective of routing is to minimize total wirelength without violating the limits of congestion and critical timing.

Differing from the traditional learning-free solvers for placement and routing, learning-based models are recently explored and applied to (macro) placement [1, 2] and routing [3], which are mostly based on reinforcement learning (RL). Like in vision, language and other domains, an end-to-end network is often welcomed for the possibility for global optimization of the whole system, while such pure neural networks for joint placement and routing remains unresolved in literature. In particular, there are two standing issues worth further study in this paper: i) the macros are of mixed-size with varying width/height [4], bringing difficulty to the discrete grid-based placement by (current reinforcement) learning; ii) the scale of grid size in routing dataset e.g. ISPD-07 [5] can be intractable for RL as currently its action has to be designed at the local grid point level [6] rather than generate the whole routing across grids in one shot. In this paper, we aim to develop a flexible (mixed-size friendly) approach. More ambitiously, we further aim to enable a pure neural pipeline with both learnable placement net and routing net. Specifically, the placement part is fulfilled by a policy gradient based RL method for macros by considering their sizes, followed with gradient-based optimization for placing stand cells. The routing part involves a conditional generative model to finish the routing across pins at net level. The highlights of this work are:

1) We propose an RL-based model for learning mixed-size macro placement, which differs from existing learning-based placers [2, 1, 7] that often consider the macro by coarse grid-based mask. As such, the placement results are more realistic and require less post-processing to resolve collision.

2) We propose a conditional generative routing network to perform routing, in one-shot for each net. In contrast, existing RL-based solvers [6] need to perform routing step by step at grid point level within each net, in a sequential and inefficient manner. Moreover, the order for nets to route is adaptively optimized by learning instead of a pre-fixed one as performed in existing routers.

3) Combining these techniques, we develop a neural network-based pipeline for placement and routing, which to our best knowledge, is the first pure neural networks for placement and routing[2].

4) Experimental results on benchmarks show the relatively cost-efficiency (compared with RL-based routing solver [6]) and competitive performance. Source code will be made publicly available.

## 2    Related Work

For the inter-discipline nature of this paper, we briefly introduce the necessary background and related work to properly position our work with different communities from EDA to machine learning. Due to page limit, classical methods for placement and routing are presented in Appendix A.1.

**Learning-based Solvers for Placement.** Machine learning has recently been introduced in placement which may help reduce the heuristics. [7] devises a cyclic framework between reinforcement learning (RL) and SA modules, in which the RL module alters the relative spatial sequence between circuit components, while SA searches the solution space based on RL initialization. Following the seminal work [1] of learning sequential decision-making for macro placement, the method called DeepPlace in [2] proposes a joint learning technique for the placement and subsequent routing via RL.

**Learning-based Solvers for Routing.** The recent work [8] presents an attention-based RL method for obtaining pin order within each net (rather than net order in this paper which is of much larger size), followed by a classical pattern router. A DQN agent [6] is developed to decide on the routing direction on a grid graph at each step. It makes up a simple 12-element vector to represent the state of the environment. However, the model is trained on synthesized $8 \times 8$ and $16 \times 16$ grids, with no more than 50 nets that consist of 2 or 3 pins. To make the learning more scalable, a two-page preliminary work [9] formulates the routing of a net as an image-to-image translation and uses a variational auto-encoder (VAE) [10] model to generate the result. However, the model is merely capable of handling a net with no more than three pins on a $64 \times 64$ grid. Compared with classic routing solvers, existing RL-based methods can be much more time-consuming, making the end-to-end learning of

---

[2]The recent work [2] also aims to achieve learning of both placement and routing, while for fast routing to achieve fast rollout for fitness evaluation, it runs a heuristic rip-up and reroute algorithm while we for the first time make the whole pipeline learnable for both placement and routing.

both placement and routing nets very difficult. Moreover, compared with [8] learning the pin order inside a net, we try to learn the order for routing at the net level which is new in literature.

**Generative Models for Placement and Routing.** There are emerging works on introducing generative models for chip design. [11] proposes a generative adversarial network (GAN) [12] guided well generation framework to mimic experts' behavior from high-quality manually-crafted layouts. [13] adopts a GAN for generating wells and guides the placement in analog circuit layout synthesis. [14] proposes a generative model for the placement optimization of analog integrated circuit basic blocks. ThermGAN [15] treats the thermal map estimation problem as an image-generation problem using the generative model. [16] uses GAN to predict the congestion heatmap to assist classical routers. The work [17] proposes a conditional GAN to solve the multi-terminal path-finding task. In [18] the generative models are adopted to synthesize diverse layout patterns. More broadly speaking, generative models have also been recently applied in combinatorial optimization, specifically via a latent space learning and search scheme as done in CVAE-Opt [19]. In this paper, we adopt cGAN to generate routes in one shot for each net by regarding the layout to be routed as an image. To our best knowledge, no generative model has been successfully devised and applied to the pin routing problem for each net, which in fact involves complicated and constrained routing.

## 3 Methodology

**Approach Overview.** Given a netlist as input, the goal of placement is to place the macros and standard cells on the chip canvas, ideally with a minimized overlapping areas. Based on the placement results, routing is performed in general to minimize the total wirelength while not violating the constraints. For placement, we aim to flexibly and efficiently handle the mixed-size macros via an RL scheme, and meanwhile optimize the net order for routing. For routing, we propose a conditional generative model to obtain the routes in one-shot instead of performing sequential pin connection as done in previous learning-based [6] as well as classic routers [20, 21]. The stacking RL placement network and generative routing network can be learned via gradient back propagation in an end-to-end manner, which meanwhile completes the whole task in line with the state-of-the-art learning-based placement and routing solvers [2, 1]. Note our work is pure learning based in contrast to [2] that still involves unlearnable classic routers in the whole pipeline. The pure neural architecture implies that our model has the potential to enjoy higher capacity by using larger model for further improvement.

### 3.1 Reinforcement Learning for Mixed-size Placement

A natural idea for classical placers to address the mixed-size issue for placement is to adopt a hierarchical approach based on partitioning. However, it sacrifices the solution quality since each sub-problem is solved independently. Meanwhile, DeepPlace as a learning-based placer assumes that each macro only occupies one cell in the grid graph and ignores pre-placed macros, which leads to a severe overlap issue in the final placement result. This motivates us to extend the formulation of DeepPlace by considering the real size of macros as well as initial placement information. We still adopt [22] as the CNN backbone and GCN [23] as the GNN backbone which consists of three layers that contain 16, 32 and 16 feature channels. The policy network is updated by Proximal Policy Optimization (PPO) [24], in line with the effective setting adopted in [2]. The revised elements of the Markov Decision Processes (MDPs) for mixed-size placement are defined as:

- **State $s_t$:** the state representation still consists of global image $I$ portrayed the layout and netlist graph $H$ contains detailed position of placed macros, except that the initial state of $I$ is no longer a zero matrix $I_{n \times n} = \mathbf{O}$. Instead, our model preprocesses the positions of fixed macros in the dataset and sets $I_{xy}$ as 1 if $(x, y)$ has already been occupied before placement.

- **Action $a_t$:** the action of RL agent is to find the central of current macro, and position $(x_o, y_o)$ is available if all points $p$ in the region $\mathbf{R}$ satisfy $I_p = 0$, where $\mathbf{R} = \{(x, y) \mid |x - x_o| \leq \frac{h}{2}, |y - y_o| \leq \frac{w}{2}\}$ and $h, w$ denote the height and width of the current macro respectively.

- **Reward $r_t$:** to further control the overlap in the final placement, the reward at the end of episode is a negative weighted sum of wirelength, routing congestion and overlapping area from the final solution: $R_E = -L_{wl} - \lambda_1 \cdot L_{cg} - \lambda_2 \cdot L_{ol}$ as weighted by $\lambda_1$ and $\lambda_2$.

### 3.2 Conditional Generative Learning for Pin Routing in Net

**The Global Routing Grid Protocol.** In global routing, the physical chip is usually divided into rectangular areas, as shown in the left part of Fig. 1. Each area is called a global routing cell (Gcell), which corresponds to a node $v_i \in V$ in the grid graph $G(V, E)$ on the right side. While each edge $e_{ij} \in E$ represents the joint boundary between abutting Gcells $v_i$ and $v_j$. For edge $e$, its capacity $c_e$ is the allowed maximum number of wires that can cross $e$, and the usage $u_e$ is the actual number of wires that $e$ has been assigned. The overflow $o_e = max(0, u_e - c_e)$ denotes the amount of wires beyond $c_e$. Each route conforms to the rectilinear Steiner tree structure [25].

**Grid-based Conditional Generative Routing.** The overall routing task is composed of numerous nets whose routing can be independent to each other. A single net consists of a series of pins placed in the nodes of the grid graph. Thus in this paper, we mainly consider the routing over multiple pins in one net. Given a single net, we formulate the one-net routing problem as the mapping from a one-net routing image $x$ to the corresponding route layout image $y$, where $x$ contains three channels: 1) the locations of pins,

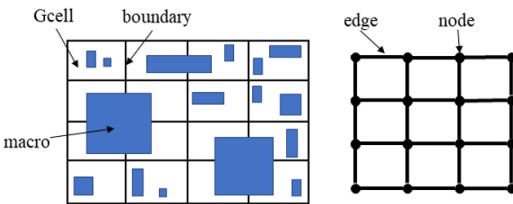

Figure 1: A chip partitioned into global cells with mixed-size macros, and its grid graph layout.

2) the availability of horizontal grid edges, and 3) the availability of vertical grid edges, and one channel for $y$. As to the output, the value of each pixel of $y$ denotes the likelihood of whether this grid point belongs to the route or not. Hence, the synthesis of routes can also be viewed as a binary classification of the pixels in $y$. The pixels whose likelihood is higher than the threshold are subsequently collected to form the route. In case of disconnected routes, we apply maze routing to refine the outcomes. In the initial design of our model, we do not adopt the random noise $z$, which mainly leads to producing fairly stochastic outputs, since the routing task barely requires stochasticity.

Our routing model adopts the conditional generative adversarial framework, which has shown effectiveness in image generation. The generator is composed of a basic generator for the input size of $64 \times 64$ or below (the smaller ones are padded to $64 \times 64$) and an extension for the input size of larger than $64 \times 64$. The discriminator consists of two sub-discriminators to estimate routes from validity and realness perspectives. We, furthermore, design an enhanced loss to improve the performance of our model. The structure of the generative model is visualized in Fig. 2.

#### 3.2.1 Layout Input-size-Adapting Generator

During routing, the physical chips are decomposed into Gcells in terms of various widths and heights as shown in Fig. 1, causing the diversity of the scale of the corresponding grid graphs. To make it more tractable, we develop an input-size-adapting generator to handle various grid graphs.

First, we construct a basic generator, $G_{base}$, to solve grids not larger than $64 \times 64$ as the chip is divided into $64 \times 64$ tiles in the macro placement stage. The architecture proposed by [26] is partly adopted as the backbone of $G_{base}$, which has been proven successful in generative tasks. Our basic generator contains four components : 1) a convolutional front-end, 2) a series of residual blocks, 3) a transposed convolutional component, and 4) a convolutional layer to generate the output.

Second, to handle larger grids, we establish another generator $G_{large}$, which is composed by two sub-networks: $G_{inner}$ and $G_{outer}$ ($G_i$ and $G_o$, for simplicity). $G_i$ and $G_o$ are termed as the guiding network and the filling network, respectively. The guiding network consists of the first three parts that $G_{base}$ owns. In contrast, the components of filling network are similar to $G_{base}$, and correspondingly we use $G_o^k (k = 1, 2, 3, 4)$ to denote them. We feed the input grid to $G_o^1$ to obtain a feature map, and downsample the input grid to feed $G_i$ to acquire another feature map. $G_o^2$ takes in the element-wise sum of these two feature maps and integrates the guiding information into $G_o$, and the hybrid feature map then is converted into the output. The architecture of our generator is illustrated in Fig. 2(left). We can further incrementally stack additional sub-networks on $G_{large}$, and model compression techniques can be used to help keep the inner network neat. While training the networks, we first pre-train the sub-networks separately, and then we jointly train them to fine-tune the whole network.

**Remarks.** The CNN-based generator coincides with routing: 2-D neighborhood structure, translation equivariance and locality. Amid routing, chips are formulated as grids, which are further transformed into images. Routing also exhibits translation equivariance since translating a whole net with the

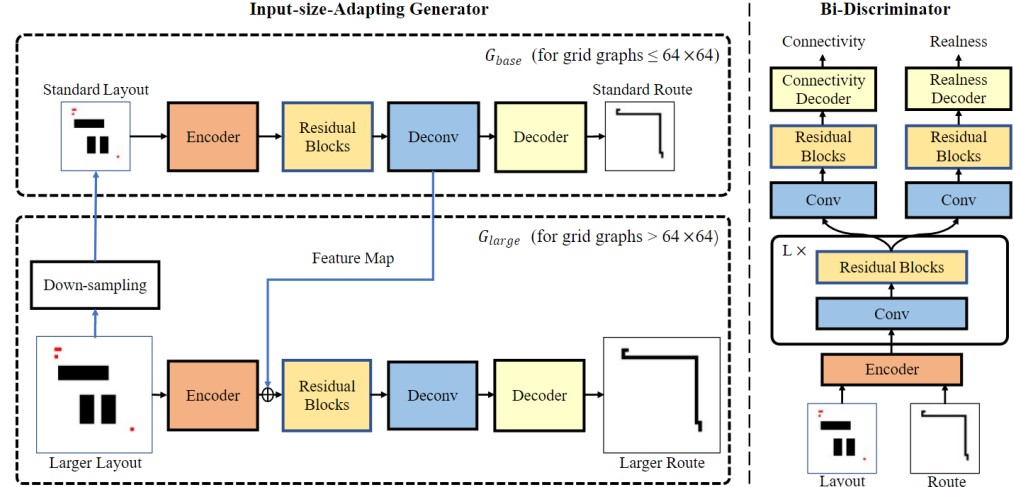

Figure 2: **Architecture of the proposed generative routing model using cGAN. Left Top:** the standard generator to handle grid graphs layout as input no bigger than 64×64. We pad black pixels to those smaller than 64×64. **Left Bottom:** the generator for larger grids. In the input layout, the red pixels represent pins, and the black blocks denote edges without capacity. We append $G_{large}$ to $G_{base}$, and the two networks are jointly trained on large grids. The element-wise sum of the feature map of $G_{large}$ and the feature map from $G_{base}$ is fed to the residual blocks of $G_{large}$ as the input. **Right:** The architecture of the bi-discriminator with two branches for connectivity and realness scoring. These two branches are trained with connectivity labels and realness labels, respectively.

context will not change the routing result. Moreover, for each grid node, the convolution kernel gathers the information from locally adjacent vertices, especially those directly connected to it, to form local routing features. The holistic route is then produced. Rather than simply stacking layers to handle long-range dependencies, the well-trained guiding network provides global information equivalent to long-range dependencies.

### 3.2.2 Bi-Discriminator to Consider both Realness and Connectivity

Routing problems have an inherent constraint that all pins should be connected. Therefore we devise a discriminator to evaluate the connectivity of the output. To effectively train the connectivity discriminator, we develop an algorithm to accurately figure out the connectivity of each fake and real route, and then we employ the results as labels. Connectivity alone is not sufficient to evaluate the authenticity of the output, so we adopt another discriminator to estimate the realness of the output, as the original discriminator in GAN. Overall, the adversarial loss of our model can be expressed as

$$\mathcal{L}_{adv}(G, D) = \sum_{i=1,2} \lambda_i \left( \mathbb{E}_{x,y} \left[ \log D_i(x, y) \right] + \mathbb{E}_x \left[ \log(1 - D_i(x, G(x))) \right] \right), \tag{1}$$

where $D_1$ and $D_2$ denote the connectivity discriminator and the realness discriminator, respectively, and $\lambda_1$ and $\lambda_2$ represent the corresponding weights s.t. $\lambda_1 + \lambda_2 = 1$. The discriminators share the convolutional front-end and a stack of $L = 3$ convolutional and residual blocks, and then they make evaluations from different angles as depicted in the right part of Fig. 2.

### 3.2.3 Enhanced Model Loss

With the cGAN objective mixed with a traditional loss, such as L1 and L2 loss, training is inefficient as most grid points are easy negatives that cannot yield effective learning signals. In addition, tons of trivial negatives impair the training and give rise to a degraded model, and the output, thus, inclines to converge to an empty route. To bridge the gap between easy negatives and scarce positives, we apply the focal loss [27] and modify it to fit our task:

$$\mathcal{L}_{FL}(G) = -\mathbb{E}_{x,y} \left[ \frac{1}{N} \sum_{i=1}^{N} \alpha \left[ y_i(1 - g_i)^\gamma \log g_i + (1 - y_i)g_i^\gamma \log(1 - g_i) \right] \right], \tag{2}$$

where $i = 1, \ldots, N$ represents grid points, and $y_i$ and $g_i$ respectively denotes the real and generated value of corresponding grid point. We also incorporate the L2 loss into our objective to approach the real routes, and because it has been found beneficial to the synthesis [28, 29].

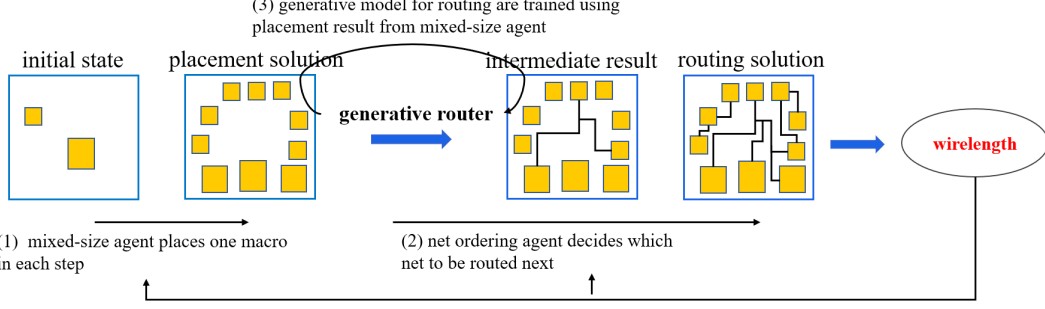

Figure 3: **Our neural macro placement and routing pipeline.** Given netlist as input, our mixed-size agent sequentially places macros on the chip layout. Generative router is then adopted to route the net chosen by net order agent. Inspired by EM algorithm, we update generative router by placement result from mixed-size agent, then placement and net order agents are learned jointly in a whole reinforcement learning framework to minimize wirelength calculated by trained generative model.

The introduction of the connectivity discriminator improves the correctness of the results, but at the same time, it may slightly increase the wirelength. Since the wirelength has an accurate theoretical lower bound, i.e. half-perimeter wirelength (HPWL) of the bounding box of a net, we take the difference between the length of the generated route and the HPWL as a regularization term to limit the wire length. We use $\mathcal{L}_r(G) = \mathbb{E}_x \left[ \| l(G(x)) - h(x) \|_1 \right]$ to represent the regularization term, where $l(G(x))$ denotes the length of the generated route, and $h(x)$ denotes the HPWL of the net.

The overall enhanced objective of our model gathers the above losses:

$$\min_G \left( \left( \max_D \mathcal{L}_{adv}(G, D) \right) + \mu_{FL} \mathcal{L}_{FL}(G) + \mu_{L2} \mathcal{L}_{L2}(G) + \mu_r \mathcal{L}_r(G) \right), \tag{3}$$

where $\mu_{FL}$, $\mu_{L2}$ and $\mu_r$ are defined as the factors of $\mathcal{L}_{FL}$, $\mathcal{L}_{L2}$ and $\mathcal{L}_r$, respectively.

### 3.3 Neural Macro Placement and Routing Pipeline

Combining the RL-based model for learning mixed-size macro placement with one-shot generative routing network to perform routing as we introduce above, we propose a pure neural pipeline for macro placement and routing. Fig. 3 shows the flow of our mixed-size macro placer with adaptive reward function between coarse HPWL estimation to wirelength from the neural router. Given the circuit information, our mixed-size agent sequentially places the macros on the chip layout, after which the generative model for routing is adopted to connect the net chosen by net order agent and finally calculate wirelength as feedback. Inspired by EM algorithm, we first update the generative router using placement result from mixed-size agent (similar to **E step**), then placement and net order agents are learned jointly in a whole reinforcement learning framework to minimize wirelength calculated by trained generative model (corresponding to **M step**) following a recursive pattern.

### 3.3.1 Reward Adaptation between Coarse HPWL and Router's WL

HPWL is a common metric for estimating the true wirelength decided by routing. In our neural pipeline, however, we apply a one-shot generative routing network to route all the nets directly, which reduces bias in the reward signal. Nevertheless, it is worth noting that the untrained policy network for placement would start with random weights so that placement results are of low quality. As a result, the distribution of pins in a single net will spread out, which is difficult for a generative model-based router to produce accurate route layout images. To tackle this problem, we propose an adaptive scheme to calculate wirelength for our placement agent, integrating HPWL and neural router's output simultaneously. We introduce $\lambda$ to scale two values and define the smoothed wirelength $WL_s$:

$$WL_s = \lambda \cdot WL_n + (1 - \lambda) \cdot HPWL \tag{4}$$

where $WL_n$ is the feedback of neural router. In each iteration, variable $\lambda$ is updated by function $1 - e^{-0.01 \cdot n_{iter}}$. Initially, $\lambda$ begins with 0 so that the wirelength is determined by $HPWL$ that serves as a coarse reward signal. As the training proceeds, the feedback of neural router gradually becomes a prominent factor to provide a more accurate objective for the placement agent.

### 3.3.2 Learning Net Order to Route

The order in which nets are routed is one of the most critical factors that affects the routing quality [30]. Most classical routers determine the net order by heuristics, e.g., routing smaller nets earlier [31] due

to the flexibility of finding free path. However, there are diverse definitions for "smaller net", none of which is proved to be optimal. What makes the situation worse, the complexity of real routing procedure requires us to change the net order dynamically, which is hard to implement in such a complicated system. Fortunately, our neural router divides the routing task into a series of one-net routing problems and then solves them independently, making it convenient to learn the net order.

We build net order learning module upon the neural router by developing a RL agent to determine which net to route next. Inspired by the structure of placement agent, the state of net order problem consists of routing image $R$ as a representation of current routing layout, and graph $G$ indicating the connectivity between nets. There are three channels of $R$: the locations of routed net, the capability of horizontal and vertical grid edges. Graph $G$ is an edge-to-vertex dual of netlist graph $H$, whose vertices denote nets (i.e., hyperedges of $H$) and edges denote common cells between nets (i.e., nodes of $H$). Note that the state of placement and net ordering tasks are quite similar while both seek to minimize the total wirelength, we combine them into a whole RL framework by adopting same policy network to generate feature embeddings for two tasks respectively. The united structure without heuristic solver reflects the strongly coupled relationship between placement and routing, which differs from [2] that merely applies router as a black box to calculate the reward.

## 4 Experiments

### 4.1 Protocols and Setup

Experiments are conducted on a server with RTX 3090 GPUs and AMD 3970X 32-Core CPU, and implemented by PyTorch. We term our whole approach for placement and routing as **PRNet**.

**Benchmarks & Datasets.** For placement, we validate our RL agent for mixed-size macro placement using ISPD-2005 benchmark [32] after pre-processing, such that most fixed macros are exchanged for movable ones in line with [2]. For routing[3], we choose the ISPD-07 [5] benchmarks to produce routing instances and use the routes generated by the strong classic router [33] as training labels.

In the ISPD-07 benchmarks, some nets can contain up to hundreds of pins, but the average amount of pins in a single net is still about 4. In other words, massive nets contain no more than 4 pins. Therefore, the routing model should have sufficient ability to route the easy nets. From around 750K routing instances, we collect 30K 64×64 nets as the Route-small-4 dataset whose instance contains up to 4 pins, 80K 64×64 samples as Route-small, and 100K 128×128 samples as Route-large to evaluate the model's ability. Each of the three datasets is randomly divided into a training set (80%) and a test set (20%). These three training sets are used to train generative routing models and pick up the best one according to their performance on the test sets. Then we continue to train the best 64×64 model with additional 200K 64×64 instances and train the 128×128 model with additional 400K 128×128 instances, which is used to perform experiments on the ISPD-98 routing benchmarks [34].

**Training.** To train the placement and net ordering RL agent, we use PPO [24] to update the policy network and Adam optimizer [35] is utilized with a learning rate of $2.5 \times 10^{-4}$. For training the routing models, we use Adam with learning rate of $2 \times 10^{-4}$, $\beta_1 = 0.5$, $\beta_2 = 0.999$ and a weight decay of 0.01. We employ a batch size of 64. A linear learning rate decay is also applied.

**Evaluation.** In mixed-size placement, we adopt HPWL as the proxy of wirelength and overlapping area to evaluate both methods while we introduce wirelength (WL) and routing congestion (RC) [36] in overall placement and routing. For routing, since there is little metric for generative routing model, we introduce the metrics correctness rate (CrrtR) and wirelength ratio (WLR) to evaluate the generated results of generative models on the datasets. CrrtR signifies the ratio of the amount of connected overflow-free routes to the number of all routes, or in short the accuracy of generated result. WLR represents the ratio of the total wirelength of connected overflow-free routes to the total wirelength of the corresponding real routes. Lower WLR indicates that the route requires fewer wires. In the experiments on the ISPD-98 routing benchmarks, wirelength, overflow and runtime are used.

### 4.2 Results on Mixed-size Placement

We compare the total wirelength together with overlapping area of our mixed-size approach with the state-of-the-art and open-sourced method called DeepPlace [2] as shown in Table 1. Both methods generate intermediate macro placement via RL, and then adopt gradient-based optimization placer

---

[3]In fact we have limited choice for constructing our deep model training dataset, as there is little public dataset for training generative models for routing, and few classical routers are open-source.

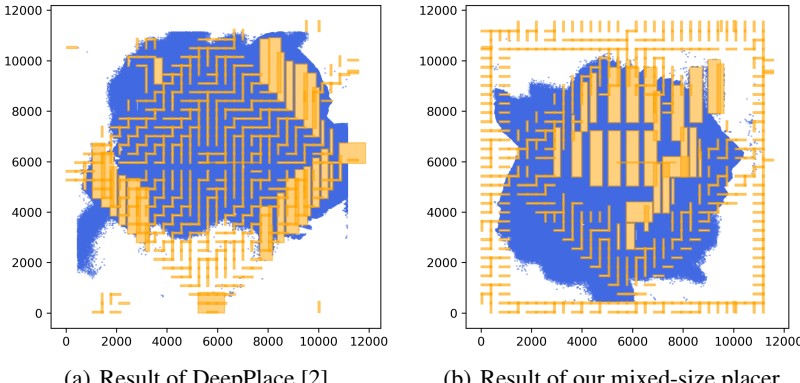

|  (a) Result of DeepPlace [2]. | (b) Result of our mixed-size placer. |

Figure 4: Visualization of macro (in orange) /standard cell (in blue) placement by DeepPlace [2] and our mixed-size placer on circuit $bigblue1$. Our placer tends to place large macros in the center of canvas to avoid overlapping, while they are close to each other on the boundary for DeepPlace.

Table 1: Comparison on mixed-size placement task on the eight circuits from ISPD-2005.

| Circuit | # Cells | # Mov. | Mixed-size technique (ours) | | DeepPlace [2] | |
| | | | Wirelength↓ | Overlap Area↓ | Wirelength↓ | Overlap Area↓ |
|---|---|---|---|---|---|---|
| adaptec1 | 211K | 514 | 82783826 | **12606828** | 80117232 | 66608273 |
| adaptec2 | 255K | 542 | 123307824 | **19485631** | 123265964 | 47085963 |
| adaptec3 | 451K | 710 | 232373680 | **58588016** | 241072304 | 140272759 |
| adaptec4 | 496K | 1309 | 234008876 | **73075220** | 236391936 | 169853555 |
| bigblue1 | 278K | 551 | 141020208 | **2041890** | 140435296 | 3519755 |
| bigblue2 | 558K | 948 | 144803296 | **70702107** | 140465488 | 103663199 |
| bigblue3 | 1097K | 1227 | 468632064 | **39664931** | 450633360 | 574956948 |
| bigblue4 | 2177K | 659 | 1001315712 | **67794270** | 951984128 | 87630042 |
| ratio | - | - | 1.000 | 1.0 | 0.987 | 3.9 |

as used in [37] to obtain complete placement solution. With only a slight increase of the total wirelength (within 1.3% difference on average), our mixed-size macro placer achieves approximately $4\times$ reduction over DeepPlace on the overlapping area, stressing the importance of modeling macro's shape in state space. Moreover, the reduced overlapping area requires less post-processing to resolve collision, which facilitates improvement of wirelength in the long term. Examples of our mixed-size placer and DeepPlace on circuit $bigblue1$ are visualized in Fig. 4.

## 4.3 Results on Routing

**Comparison of Generative Backbones.** We compare our model with a CVAE based router using CNN as the backbone [9] and use CVAE*(CNN) to denote it. We then combine the CVAE*(CNN) with a vanilla discriminator and our bi-discriminator to build CVAE*-GAN and CVAE*-bcGAN, following the idea of [38]. In addition, we implement a U-Net based cGAN following pix2pix [29] and use cGAN(U-Net*) to denote it. Then we remove the discriminator to obtain a U-Net [39] generator and define it as U-Net*. We further replace the discriminator of cGAN(U-Net*) with our bi-discriminator. We also try to train the RL agent following the work of [6], but it fails to converge after 2-week training. Table 2 shows the comparison results. The ResNet-based [40] models outperform the counterparts based on CVAE*(CNN) and U-Net*. Our model achieves approximately $2\times$ correctness rate, 14.7% improvement of the wirelength on Route-small-4 and 2.4% on Route-small over CVAE*(CNN). The vanilla cGAN discriminator slightly improves CVAE*(CNN) and U-Net* on one side while sacrificing the other side, and it debases the ResNet generator. However, the bi-discriminator strengthens the generators except for the CVAE*(CNN).

**Ablation Studies.** We conduct ablation experiments to investigate the contributions of the design choices in our model. In Table 2, we compare the full version with ResNet-based cGAN, as well as the pure ResNet generator. The ResNet generator outdoes the cGAN, but the bi-discriminator significantly improves the generator. Moreover, the enhanced loss improves the wirelength at the marginal expense of correctness. Appendix A.3.1 contains further details of comparison among loss functions, and Appendix A.3.2 shows the effectiveness of the input-size-adapting network.

Table 2: Evaluation of different backbones w.r.t. correctness rate (CrrtR) and wirelength ratio (WLR) for the routing on: Route-small-4 and Route-small. cGAN: the vanilla cGAN model with a single realness discriminator; bcGAN: the bi-discriminator version. EL: enhanced loss in Eq. 3.

| our router w/ different generative models | Route-small-4 | | Route-small | |
|---|---|---|---|---|
| | CrrtR↑ | WLR↓ | CrrtR↑ | WLR↓ |
| CVAE*(CNN) [9] | $0.414_{\pm0.020}$ | $1.179_{\pm0.033}$ | $0.397_{\pm0.008}$ | $1.042_{\pm0.006}$ |
| CVAE*-cGAN(CNN) | $0.557_{\pm0.065}$ | $1.292_{\pm0.108}$ | $0.439_{\pm0.021}$ | $1.315_{\pm0.015}$ |
| CVAE*-bcGAN(CNN) | $0.474_{\pm0.048}$ | $1.525_{\pm0.029}$ | $0.488_{\pm0.007}$ | $1.241_{\pm0.012}$ |
| U-Net* [39] | $0.724_{\pm0.001}$ | $3.306_{\pm0.266}$ | $0.524_{\pm0.005}$ | $1.232_{\pm0.016}$ |
| cGAN(U-Net*) [29] | $0.602_{\pm0.009}$ | $1.028_{\pm0.001}$ | $0.532_{\pm0.011}$ | $1.286_{\pm0.022}$ |
| bcGAN(U-Net*) | $0.721_{\pm0.012}$ | $1.134_{\pm0.055}$ | $0.552_{\pm0.007}$ | $1.104_{\pm0.054}$ |
| ResNet [40] | $0.783_{\pm0.002}$ | $1.023_{\pm0.003}$ | $0.594_{\pm0.004}$ | $1.030_{\pm0.007}$ |
| cGAN(ResNet) | $0.698_{\pm0.010}$ | $1.073_{\pm0.011}$ | $0.568_{\pm0.020}$ | $1.320_{\pm0.151}$ |
| bcGAN(ResNet) | $0.804_{\pm0.021}$ | $1.035_{\pm0.013}$ | $\mathbf{0.738}_{\pm0.005}$ | $1.036_{\pm0.002}$ |
| bcGAN(ResNet)+EL (full version of our router) | $\mathbf{0.814}_{\pm0.001}$ | $\mathbf{1.010}_{\pm0.000}$ | $0.735_{\pm0.010}$ | $\mathbf{1.018}_{\pm0.004}$ |

Table 3: Evaluation of wirelength (WL) and routing congestion (RC) for overall placement and routing pipeline on ISPD-05 benchmark. "GR": our generative router; "NOL": net order learning.

| Circuits | RL-based Placer (i.e. DeepPlace [2]) | | RL-based Placer + GR | | RL-based Placer + GR + NOL (our PRNet) | |
|---|---|---|---|---|---|---|
| | WL↓ | RC↓ | WL↓ | RC↓ | WL↓ | RC↓ |
| adaptec1 | 6149 | 10.565 | 5940 | 10.464 | **5787** | **9.386** |
| adaptec2 | 23659 | 46.278 | 23048 | 45.249 | **22977** | **35.504** |
| adaptec3 | 30154 | 62.751 | 29711 | 73.324 | **29462** | **43.207** |
| adaptec4 | 47933 | 128.257 | 47406 | 121.435 | **46964** | **65.796** |
| bigblue1 | 7634 | 11.480 | 7385 | 12.391 | **7230** | **11.289** |
| bigblue2 | 16775 | 26.318 | 16693 | 45.945 | **16617** | **25.498** |
| bigblue3 | 42550 | 67.124 | 41617 | 70.187 | **40509** | **64.964** |
| bigblue4 | 15847 | 34.903 | 15356 | 42.096 | **15283** | **24.223** |

Table 4: Evaluation of wirelength (WL) and runtime in seconds (Time) with three classical routers on ISPD-98 routing benchmarks. Note that the overflow (OF) is all zero for all methods.

| Circuits | Our router | | NTHU-Route 2.0 [30] | | BoxRouter 2.0 [41] | | FastRoute 3.0 [42] | |
|---|---|---|---|---|---|---|---|---|
| | WL↓ | Time(s)↓ | WL↓ | Time(s)↓ | WL↓ | Time(s)↓ | WL↓ | Time(s)↓ |
| ibm01 | **62337** | 59.2 | 62498 | 1.54 | 62659 | 33 | 64221 | 0.64 |
| ibm02 | 170270 | 179.9 | **169881** | 3.15 | 171110 | 36 | 172223 | 0.85 |
| ibm03 | **146362** | 194.6 | 146458 | 1.49 | 146634 | 18 | 146753 | 0.49 |
| ibm04 | **165874** | 254.4 | 166452 | 3.81 | 167275 | 116 | 170146 | 2.7 |

**Test Results.** We test our conditional generative router on the ISPD-98 benchmarks and compare the wirelength, overflow and runtime with three classical routers that perform best on ISPD-98 benchmarks. Table 4 shows the results. Our generative routing model presents competitive consequences on wirelength, while it takes a longer time to accomplish the routing task, compared with strong heuristic baselines [30, 41, 42]. Our model takes an image encoded from the whole grid with a net as the input and sequentially solve each net in a one-shot manner, while classical routers only consider the local area, which may obtain fewer wirelength yet consume more time. However, it is still much more efficient and easier to train than the RL-based router [6] in our empirical experience.

### 4.4 Comparison with a Concurrent Version of Our Router and Other Classic Routers

By default, our generative router routes the nets sequentially (i.e. one net by one net) such that it produces routes with lower wirelength and overflow. In another word, the batch size in the inference stage for the deep routing network is 1 which is not common and does not release the computing advantage of GPU. To speed up our generative router, we set the batch size to 16 in the inference stage to enable our router to run in a concurrent manner.

However, the concurrent version may degenerate the performance of the outputs on wirelength and overflow. As illustrated in Fig.1 in Appendix, the bounding box of $net1$ overlaps with that of $net2$, and the bounding box of $net3$ is far away from them. When routing these three nets concurrently, the routes of $net1$ and $net2$ may intersect with each other and cause conflict in the usage of routing capacity, which leads to overflow and makes routing harder for the rest of nets.

We first test our conditional generative routing model with each net routed sequentially (i.e. a batch size of 1) on the ISPD-98 benchmarks. In addition, we run our generative router with nets routed concurrently (a batch size of 16), which may trade wirelength and overflow for runtime, and we reimplement the post-processing module by C++. The sequential and concurrent variants of our

Table 5: Evaluation of wirelength (WL), overflow (OF) and runtime (Time) with three classical routers on ISPD-98 routing benchmarks. Sequential denotes the variant of our router sequentially routing each net, while Concurrent represents the concurrently routing variant. Numbers including WL, OF and time for the compared router are directly quoted from the paper BoxRouter [45] where the hardware setting is 2.8 GHz Pentium-4 Linux machine with 2G RAM. Note $ibm05$ is a trivial case as also pointed out in [45] due to the sufficient routing resources.

| Circuits | Our router (Sequential) | | | Our router (Concurrent) | | | Labyrinth 1.1 [43] | | | Fengshui 5.1 [44] | | | BoxRouter [45] | | |
|---|---|---|---|---|---|---|---|---|---|---|---|---|---|---|---|
| | WL↓ | OF↓ | Time(s)↓ | WL↓ | OF↓ | Time(s)↓ | WL↓ | OF↓ | Time(s)↓ | WL↓ | OF↓ | Time(s)↓ | WL↓ | OF↓ | Time(s)↓ |
| ibm01 | **62337** | 0 | 59.2 | 64094 | 43 | 17.6 | 75909 | 340 | 4.46 | 66006 | 189 | 15.1 | 65588 | 102 | 8.3 |
| ibm02 | **170270** | 0 | 179.9 | 173391 | 25 | 27.1 | 201286 | 371 | 7.34 | 178892 | 64 | 47.9 | 178759 | 33 | 34.1 |
| ibm03 | **146362** | 0 | 194.6 | 148629 | 5 | 24.4 | 187345 | 258 | 8.66 | 152392 | 10 | 35.2 | 151299 | 0 | 16.9 |
| ibm04 | **165874** | 0 | 254.4 | 168897 | 142 | 36.1 | 195856 | 947 | 21.88 | 173241 | 465 | 54.1 | 173289 | 309 | 23.9 |
| ibm05 | **408421** | 0 | 189.9 | 409356 | 4 | 43.7 | 420581 | 0 | 5.35 | 412197 | 0 | 104.8 | 409747 | 0 | 49.5 |
| ibm06 | **278029** | 0 | 232.9 | 279664 | 27 | 38.2 | 341618 | 495 | 14.04 | 289276 | 35 | 80.1 | 282325 | 0 | 33.0 |

generative routing model are defined as Sequential and Concurrent, respectively. We further compare the wirelength, overflow and runtime with three more classical routers, Labyrinth 1.1 [43], Fengshui 5.1 [44] and BoxRouter [45] in Table 5. The results of three classical routers are directly quoted from the paper BoxRouter [45]. The experiments of classical routers are performed on a 1.4GHz PentiumIII workstation with Linux operating system, and our generative routers are performed on a server with RTX 3090 GPUs and AMD 3970X 32-Core CPU. We also secure the source code of Labyrinth 1.1 and run on our machine where results are shown in Table 4 in Appendix, while the other two routers are not open-source and are difficult to reproduce.

Table 5 shows the results. On the one hand, our sequential generative routing model outperforms the classical routers on wirelength and overflow, while it takes a longer time, compared with strong heuristic baselines. On the other hand, our concurrent version dramatically speeds up the running ($3.36 \sim 7.98\times$ speedup) of generative routing model at the expense of total wirelength and overflow, and it still performs better than the three classical routers on wirelength and overflow, except the overflow on $ibm05$ and the overflow of BoxRouter on $ibm03$ and $ibm06$. Modifying our algorithms and finding other techniques to cut down the runtime are still being explored.

## 4.5 Results of Overall Placement and Routing with Ablation Study on Net Order Learning

We compare our PRNetwith DeepPlace, along with an ablation study to verify the impact of net order learning. The circuits used for evaluation are the same as in mixed-size placement, and we concentrate on macros only for simplicity. Note that the real shape of macros is ignored and the grid-based mask is coarser in [2], hence the results shown in Table 3 are not identical to those in the original paper [2]. For all test cases, our neural placement and routing pipeline outperforms the other two methods in terms of both wirelength (WL) and routing congestion (RC). The significant difference in routing congestion without net order learning indicates that net order agent is able to arrange the sequence of routing efficiently, especially on circuits $adaptec3$ and $adaptec4$. As a result, it is easy for every net to find free routing path while keeping away from congested area. In addition, training placement model with generative neural router in an end-to-end manner further improves the final wirelength, despite a little degradation of routing congestion if we discard the net order agent.

## 5 Conclusion and Outlook

We have presented a neural mixed-size placement and routing pipeline. The routing is achieved by one-shot generation of the whole path, with our devised net order learning module to dynamically adjust the routing order. Experimental results show the effectiveness of our approach.

**Limitation & potential negative impact.** The conditional generative routing model is currently trained on the routes generated by the state-of-the-art classical router. Therefore, its performance may be limited by the trainer. Unsupervised/semi-supervised learning is possible alternatives. For its negative side, AI-based placement and routing may cause the lose of jobs in EDA industry.

## Acknowledgement

This work was partly supported by National Key Research and Development Program of China (2020AAA0107600), National Natural Science Foundation of China (61972250, 72061127003), and Shanghai Municipal Science and Technology (Major) Project (2021SHZDZX0102, 22511105100).

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
