# OpenReview forum: "The Policy-gradient Placement and Generative Routing Neural Networks for Chip Design"
_NeurIPS.cc/2022/Conference — NeurIPS 2022 Accept_

### Official Review · Reviewer_wLLm · 2022-07-09

**Rating:** 6
**Confidence:** 3
**Soundness:** 3 good
**Presentation:** 3 good
**Contribution:** 3 good

**Summary:**

This paper proposed a neural network-based pipeline for placement and routing of macros and standard cells in chip design. For placement, this paper proposed an RL-based model for learning mixed-size macro placement to reduce frequent collisions in previous methods. For routing, this paper proposed a conditional generative routing network with adaptive routing order to improve previous methods with fixed routing order. Experiments show that the proposed approach has lower overlap area, smaller wire-length and lower routing congestion than previous methods.

**Questions:**

1. Why incorporating macro size into the state can bring a significant reduction of overlapping areas?
2. Does the fine grained state information in placer requires higher training costs?
3. Due to the combinatorial nature of this problem, what is the scalability of the proposed approach?
4. An estimate of the maximal number of circuit components can be tackled in a reasonable amount of time?

**Limitations:**

Yes.

**Strengths And Weaknesses:**

Strengths:
+ The chip design problem is an important and challenging one.
+ The proposed approach is technically sound and brings performance improvement over previous approaches.

Weaknesses:
- The main novelty of the proposed approach is at the routing aspect. The extension of placement module over DeepPlace is very subtle by adopting a different state definition, and it's not clear how such minor extension leads to better performance observed in experiments.
- Most evaluations of the proposed approach are divided into placement module and routing module as if two separate approaches are proposed. As the main targeted contribution is a joint design, I suggest to perform more evaluations of the proposed approach as a whole to compare with the best combination of existing placers and routers.
- Error bars are not added for Table 1, Table 3 and Table 4.

---

> ### Author Response · Authors · 2022-08-02
> **Rebuttal to Reviewer wLLm**
>
> Thank you for the detailed feedback and your recognition to our work's novelty, technical soundness and empirical strong performance. Below we respond to your comments.
>
> **Responses to the main concerns:**
>
> **Q1. *"The main novelty of the proposed approach is at the routing aspect. The extension of placement module over DeepPlace is very subtle by adopting a different state definition, and it's not clear how such minor extension leads to better performance observed in experiments.***
>
> **Re**: Thanks for your recognition to the novelty of our generative routing model. While we also state in our contribution (see line 54-56) that we also devise a novel mixed-size macro placement learning approach and show its advantage over [1] (see Table 1). In contrast, existing mixed-size placement methods are almost handcrafted heuristic methods instead of learning-based techniques (see [53, 54] in the reference list). In fact, this is ignored in [1]. The reason why incorporating macro size into the state space can bring a significant reduction of overlapping areas is explained by our response below (for the first question).
>
> [1] R. Cheng and J. Yan, “On joint learning for solving placement and routing in chip design,” NeurIPS, 2021.
>
> **Q2. *"Most evaluations of the proposed approach are divided into placement module and routing module as if two separate approaches are proposed. As the main targeted contribution is a joint design, I suggest to perform more evaluations of the proposed approach as a whole to compare with the best combination of existing placers and routers.***
>
> **Re**: Thanks for your recognition to the evaluations of our placement module and routing module. While we also investigate our proposed joint neural macro placement and routing pipeline in section 4.4 to validate the effectiveness of net order learning as well as the joint training pipeline. The significant difference in routing congestion without net order learning indicates that net order agent is able to arrange the sequence of routing efficiently, and training placement model with generative neural router in an end-to-end manner further improves the final wirelength (see line 337-342). Since our experiments only consider placement and routing of macros, the only available baseline in literature is DeepPlace as shown in Table 3.
>
> **Q3.*"Error bars are not added for Table 1, Table 3 and Table 4."***
>
> **Re**: When it comes to experiments for placement, we train the RL agent on a specific chip and then obtain the result on the same chip. Thus, there is no random partitioning of data into training and test sets. Moreover, the transition function in our Markov decision process is deterministic, which further removes randomness from the training process. As a result, we do not report error bars for Table 1, Table 3 and Table 4.
>
> **Responses to the questions:**
>
> **Q1. *"Why incorporating macro size into the state can bring a significant reduction of overlapping areas?"***
>
> **Re**: In DeepPlace (and also other previous learning-based models for macro placement) an assumption is made that the size of all macros is $1 \times 1$ in a virtual canvas (the grid size $n$ is a hyperparameter, e.g., $n = 64$). Once all macros have been placed, we map their positions to the real canvas and perform standard cell placement. This assumption is unreasonable in reality since there are various shapes of macros and the largest one may be a thousand times larger than the smallest one. As a result, DeepPlace suffers from a severe overlap issue (see Figure 7(a) in Appendix). By incorporating macro size into the state space and making the corresponding restriction on the action space, our mixed-size placer eliminates the overlapping problem (see Figure 7(b) in Appendix).
>
> **Q2. *"Does the fine grained state information in placer requires higher training costs?"***
>
> **Re**: No. Actually the runtime of RL agent for macro placement only occupies a small percentage of runtime for overall placement (including macro placement and standard cell placement). In addition, the CNN backbone in the policy network (see line 117) is quite simple so that larger size of input has little effect on runtime. Therefore, there has not been a clear improvement in training costs for finer-grained state information.
>
> **Q3. *"Due to the combinatorial nature of this problem, what is the scalability of the proposed approach?"***
>
> **Re**: We have validated our method on the very popular benchmarks ISPD-2005 containing all 8 circuits (up to 1300 macros and 2M std cells) which are directly derived from industrial ASIC designs. Therefore, we believe the scale is similar to problems encountered in normal chip design. Furthermore, the training time in all experiments is under twelve hours, so there is still potential for solving larger problems by extending the training time. For routing, the concurrent version of our generative router is even quicker than traditional routers (see Table 9 in Appendix).

---

> > ### Author Response · Authors · 2022-08-02
> > **Rebuttal to Reviewer wLLm (Cont.)**
> >
> > **Q4. *"An estimate of the maximal number of circuit components can be tackled in a reasonable amount of time?"***
> >
> > **Re**: We have tackled circuits consisting of thousands of macros and two million standard cells under twelve hours of training (see Table 1). This number can be further maximized if more time is allowed for training.

---

### Official Review · Reviewer_NDGm · 2022-07-10

**Rating:** 8
**Confidence:** 4
**Soundness:** 3 good
**Presentation:** 3 good
**Contribution:** 4 excellent

**Summary:**

This paper proposes a pure neural pipeline which is the first (to my knowledge) joint placement and routing network in literature without involving traditional heuristic solvers. Specially, a conditional generative routing network is utilized to perform routing in one shot and placement is performed by a policy gradient based RL method for macros while considering their sizes. To tackle the net order issue during routing, a net order agent is proposed to decide which net to be routed next. Experimental results on both placement and routing benchmarks show that the proposed joint networks outperform traditional separate pipeline, as well as those network with heuristic solvers.

**Questions:**

1. Why is the runtime of generative router notably slower than classical routers? Is there any way to improve the efficiency of the generative router?

**Limitations:**

The limitation is about the speed for generative routing (see my question part).

**Strengths And Weaknesses:**

Strengths:
+ This is an interesting work at the intersection between machine learning and EDA. I applaud the authors’ courage and efforts in this direction. The paper is clearly motivated and well written with informative background and technical description thus I believe both communities can learn from this work and this is a promising area worth for further efforts.
+ The most impressive novelty lies in the achievement of the first pure neural pipeline for placement and routing owing to a critical design of generative routing. This is contrast to the previous work [Cheng and Yan NeurIPS’21] that adopts traditional routing solvers in network.
+ There are additional techniques proposed in this paper, including handling the mixed-size macro and adaptive ordering of the net routing. The combination of these additional techniques makes the proposed network more effective.
+ Experimental results are relatively comprehensive and well delivered. The performance is convincing on the public benchmark ISPD 2005.

Weaknesses:
- Despite the shown better performance in terms of wirelength, the proposed generative routing model has much longer runtime compared with classical heuristic solvers. I can image and understand that the classic heuristic solvers can be very fast by tailored and industry-level optimized C/C++ codes. The authors shall better discuss or provide more results on this weakness.

Overall this paper has obvious merits regarding with novelty, and a weakness in terms of the routing speed. I am personally tolerant to the weakness because 1) deep learning for placement and routing is still in its infancy and there is still much space to improve the efficiency especially considering the classic routing algorithms are often carefully optimized using C/C++ at industry level; 2) generative models may be a promising direction (in one shot) for incorporating machine learning to the routing problem as the current idea of using reinforcement learning for routing can be even more unaffordable. Hence, I vote for accepting this paper.

---

> ### Author Response · Authors · 2022-08-02
> **Rebuttal to Reviewer NDGm**
>
> Thank you for the detailed feedback and your recognition to our work for its potential influence, technical novelty and soundness, as well as empirical strong performance. Below we respond to your comments and look foward to your feedback.
>
> **Responses to the questions:**
>
> **Q1. *"Why is the runtime of generative router notably slower than classical routers? Is there any way to improve the efficiency of the generative router?"***
>
> **Re**: Thanks for your valuable questions. Below we give potential reasons for the runtime gap. On the one hand, as you stated, compared to our method being the first endeavor to route through pure ML algorithms, the classical heuristic routers have been developed for a long period and are industry-level optimized. Also, classical routers are generally implemented by C/C++, which executes vastly faster than Python. On the other hand, we deconstructed the runtime of our code and discover that it takes massive time to transfer tensors between CPUs and GPUs. While for the main processing logic, the post-processing module, which is also implemented by Python in our router and runs on CPUs, accounts for the majority of the runtime. Moreover, to ensure the quality of routing results, we set $batchsize = 1$ in the experiments in Test Results of Section 4.3 such that each net is sequentially processed, which sacrifices the acceleration of GPUs.
>
> To improve the efficiency of the router, we have made the following efforts: 1) we re-implement the post-processing code with C++ and integrate it into the routing pipeline; 2) we re-organize the data format and data location to minimize device migration times and data format conversion times. 3) we set batchsize as 16 through parameter tuning to achieve a better runtime performance. We evaluate this version of the router with the same experimental settings as Table 4 in the main paper, and the preliminary results are presented in Table 9 in Appendix A.5.5.

---

### Official Review · Reviewer_YfL9 · 2022-07-10

**Rating:** 6
**Confidence:** 4
**Soundness:** 2 fair
**Presentation:** 3 good
**Contribution:** 2 fair

**Summary:**

The authors propose a pipeline of chip placement and routing based on machine learning methods. The pipeline consists of the following parts.
1. An enhanced RL method for macro placement
2. A conditional generative network for routing
3. A net order learning method

**Questions:**

1. The authors claim that "the method is the first joint placement and routing network without involving any traditional heuristic solver" in the abstract. What are traditional heuristic solvers? I think there still exist heuristic methods in optimization-based standard cell placement (e.g., DREAMPlace), legalization, and detailed placement. The claim may be too strong. Moreover, I think the overall enhanced objective shown in Equation (3) still belongs to heuristic methods since it lacks of strict proof and does not guarantee solution quality.
2. Table 4 only shows the results of 4 benchmarks. I do not find other results in appendix. Could the authors conduct experiments on all the benchmarks and report the results in the main body or appendix?
3. Similarly, Table 3 only shows the results of 2 benchmarks. What about other benchmarks? If the authors do not report them, readers may think that the results are not ideal.
4. The proposed routing method is "one-shot". I am confused by this term. Could the authors explain why they use this term?

**Limitations:**

The authors discuss the limitation and the social impact in A.2 and A.3, which is a great discussion.

**Strengths And Weaknesses:**

Strengths
1. The paper is well organized and developed.
2. The method and experiments are reasonable and sound.

Weaknesses
1. There are too many hyperparameters in the proposed methods. It may be difficult for others to employ this method. What is the robustness of the method? How sensitive is the method if we modify these hyperparameters?
2. The results in Table 1 are mixed. Namely, DeepPlace and the proposed method both present Pareto optimal results. It is better to show that the proposed method can achieve lower wirelength and smaller overlap at the same time.
3. In Lines 108, and 109, the authors mention that "the pure neural architecture of our model implies our model has the potential to enjoy higher capacity by using a larger model for further improvement". This claim should be discussed in detail and verified in experiments.

---

> ### Author Response · Authors · 2022-08-02
> **Rebuttal to Reviewer YfL9**
>
> We are thankful for your recogniton to our pure neural placement and routing pipeline, the technical soundness, and overall writing. Below we respond to your specific comments and we will clarify the points in our new version.
>
> **Responses to the main concerns:**
>
> **Q1. *"There are too many hyperparameters in the proposed methods. It may be difficult for others to employ this method. What is the robustness of the method? How sensitive is the method if we modify these hyperparameters?"***
>
> **Re**: To be honest, it is not a surprise that a system considering practical placement and routing can involve many hyperparameters because there are many tradeoffs between the key indicators in real-world P&R: e.g. the wire length and overflow, congestion etc.  There are hyperparameters in both placement and routing tasks, we will give a detailed explanation for each of them.
>
> For placement, hyperparameters $\lambda_1$ and $\lambda_2$ (see line 130) are adopted to balance the wirelength metric with routing congestion and overlapping area. Wirelength is the direct objective to optimize in ISPD-2005, while routing congestion and overlapping area are correlated with routability of the final placement result. Hence we set $\lambda_1 = \lambda_2 = 0.1$ in our experiments. However, the relative importance between wirelength and routability can be dynamically adjusted according to the settings of a given chip.
>
> For routing, hyperparameters $\lambda_1$ and $\lambda_2$ (see line 199) are set to adjust the weights of realness and connectivity discriminators, and $\mu_{L2}$, $\mu_{FL}$ and $\mu_r$ (see line 218) are adopted to balance the losses from L2 loss function, focal loss function and the regularization term that represents the gap between generated wirelength and the theoretical lower bound of wirelength. Intuitively speaking, we set $\lambda_1 = \lambda_2 = 0.5$ in our experiments (see line 589 in Appendix A.5.1). Routing result images typically contain a large range of blank pixels, which leads to the fact that the losses can be diluted. In addition, the difference between routing wirelength and theoretical lower bound of wirelength can be huge, we adopt a hyperparameter to constrain its order of magnitude. Therefore, we set $\mu_{L2} = \mu_{FL} = 5\times10^3$ and $\mu_r = 0.1$ in experiments (see line 591 in Appendix A.5.1). Furthermore, we compare the effects of different choices of hyperparameters, and the results are shown in the table below.
>
> |      | $\lambda_1$ | $\lambda_2$ | $\mu_{L2}$    | $\mu_{FL}$    | $\mu_r$ | CrrtR$\uparrow$ | WLR$\downarrow$ |
> | ---- | ----------- | ----------- | ------------- | ------------- | ------- | --------------- | --------------- |
> | base | 0.5         | 0.5         | $5\times10^3$ | $5\times10^3$ | 0.1     | 0.795           | 1.007           |
> |      |             |             |               |               | 1       | 0.801           | 1.011           |
> |      |             |             |               |               | 10      | 0.785           | 1.008           |
> |      | 0.2         | 0.8         |               |               |         | **0.807**       | 1.008           |
> |      | 0.2         | 0.8         |               |               | 1       | 0.802           | 1.013           |
> |      | 0.2         | 0.8         |               |               | 10      | 0.797           | 1.008           |
> |      | 0.8         | 0.2         |               |               |         | 0.759           | 1.006           |
> |      | 0.8         | 0.2         |               |               | 1       | 0.735           | **1.005**       |
> |      | 0.8         | 0.2         |               |               | 10      | 0.742           | 1.008           |
>
>
> **Q2. *"The results in Table 1 are mixed. Namely, DeepPlace and the proposed method both present Pareto optimal results. It is better to show that the proposed method can achieve lower wirelength and smaller overlap at the same time."***
>
> **Re**: We have given an explantion at line 298-302 in the paper and will make it more clear in the new version. First of all, two evaluation metrics for placement (i.e., overlap and wirelength) are mutually contradictory. For example, we can place macros as close as possible so that the wirelength is 0 while the overlap is extremely large. The motivation of our mixed-size macro placer is to resolve the severe overlap issue in DeepPlace which makes their placement results impractical for following routing tasks. Thus a slight increase in wirelength is inevitable for a trade-off. It is unreasonable to achieve lower wirelength and smaller overlap at the same time. Moreover, in Table 1 we adopt HPWL as the proxy of wirelength, but what matters most is the final wirelength after routing. As DeepPlace requires post-processing during detailed placement to resolve the collision, our mixed-size placer can achieve lower wirelength in the long term, especially for the final wirelength when routing completes.

---

> > ### Author Response · Authors · 2022-08-02
> > **Rebuttal to Reviewer YfL9 (Cont.)**
> >
> > **Q3. *"In Lines 108, and 109, the authors mention that "the pure neural architecture of our model implies our model has the potential to enjoy higher capacity by using a larger model for further improvement". This claim should be discussed in detail and verified in experiments."***
> >
> > **Re**: Thanks for your interest to this specific observation made by us. Here we provide further interpretation and also will add them to the main paper.
> >
> > Our meaning is that replacing the classic solvers with learnable models especially for the routing part is new and can naturally allow the adoption of different models of different sizes. For circuits of larger sizes, we design larger models, which utilize small generative networks as the inner network to guide global routing and combine the inner network with an outer network to accomplish the routing task (see Section 3.2.1),  to deal with them. Here we provide experiments of larger generative routing models on the dataset Route-large that contains 100k $128\times128$ instances to show the effectiveness of larger models in Table 6 in Appendix A.5.2.
> >
> > **Responses to the questions:**
> >
> > **Q1. *"The authors claim that "the method is the first joint placement and routing network without involving any traditional heuristic solver" in the abstract. What are traditional heuristic solvers?  I think there still exist heuristic methods in optimization-based standard cell placement (e.g., DREAMPlace), legalization, and detailed placement. The claim may be too strong."***
> >
> > **Re**: The traditional heuristic solver here means traditional (or classic) methods for solving placement and routing as we described in Appendix A.1. As a result, "the first joint placement and routing network without involving any traditional heuristic solver" just refers to the neural macro placement and routing pipeline in section 3.3 where standard cell placement, legalization, and detailed placement are not included.
> >
> > **Q2. *"Table 4 only shows the results of 4 benchmarks. I do not find other results in appendix. Could the authors conduct experiments on all the benchmarks and report the results in the main body or appendix?"***
> >
> > **Re**: We add additional experimental results on another 2 circuits from ISPD-98 benchmarks, and the evaluation of wirelength (WL) and runtime (Time) is listed in the table below as an extension to Table 4. The classical routers don't provide the results of ibm05.
> >
> > | Circuits | Our router |       | NTHU-Route 2.0 |      | BoxRouter 2.0 |      | FastRoute 3.0 |      |
> > | -------- | ---------- | ----- | -------------- | ---- | ------------- | ---- | ------------- | ---- |
> > |          | WL         | Time  | WL             | Time | WL            | Time | WL            | Time |
> > | ibm05    | **408421**     | 189.9 | -              | -    | -             | -    | -             | -    |
> > | ibm06    | 277829     | 232.9 | **277696**         | 3.16 | 277913        | 47   | 279471        | 1.15 |
> >
> > **Q3. *"Similarly, Table 3 only shows the results of 2 benchmarks. What about other benchmarks? If the authors do not report them, readers may think that the results are not ideal."***
> >
> > **Re**: We add new results of 4 additional circuits from the ISPD-2005 dataset. So far we have tested 6 out of 8 chips in ISPD-2005, and the remaining 2 chips will also be provided in the next few days if permitted. Additional evaluation of wirelength (WL) and routing congestion (RC) for overall placement and routing pipeline is shown in the table below. "GR" stands for our generative router and "NOL" stands for net order learning.
> >
> > |                                                     | adaptec2           |                    | bigblue1           |                    | bigblue2 |   | bigblue4  |   |
> > |-----------------------------------------------------|--------------------|--------------------|--------------------|--------------------|----------|---|---|---|
> > | variants of our PRNet                               | **WL$\downarrow$** | **RC$\downarrow$** | **WL$\downarrow$** | **RC$\downarrow$** | **WL$\downarrow$** | **RC$\downarrow$**| **WL$\downarrow$** | **RC$\downarrow$** |
> > | RL-based Placer (i.e. DeepPlace [1])                |          23659        |  46.278  | 7634               | 11.480             |   16775       | 26.318  | 15847  | 34.903  |
> > | RL-based Placer + GR                                |         23048         |       45.249           | 7385               | 12.391             |     16693     | 45.945 | 15356  | 42.096  |
> > | RL-based Placer + GR + NOL (full version of PRNet ) |         **22977**         | **35.504**   | **7230**               | **11.289**                 |   **16617**       | **25.498**  | **15283**  | **24.223**  |
> >
> > [1] R. Cheng and J. Yan, “On joint learning for solving placement and routing in chip design,” NeurIPS, 2021.

---

> > > ### Author Response · Authors · 2022-08-02
> > > **Rebuttal to Reviewer YfL9 (Cont.)**
> > >
> > > **Q4. *"The proposed routing method is "one-shot". I am confused by this term. Could the authors explain why they use this term?"***
> > >
> > > **Re**: "One-shot" refers to the pattern that our routing method can route a whole single net at a time without splitting them into subnets. In the workflow of traditional heuristic routers (see [30, 41] in the reference list), they first decompose those nets consisting of multiple pins into numerous 2-pin subnets (a 2-pin subnets only consists of 2 pins) using a rectilinear Steiner tree construction algorithm. This phase is called net decomposition. Then they solve these 2-pin nets separately, which typically degrades the routing results. In contrast, our routing method skips the net decomposition stage and directly routes the nets.

---

### Official Review · Reviewer_KgYJ · 2022-07-10

**Rating:** 4
**Confidence:** 5
**Soundness:** 2 fair
**Presentation:** 2 fair
**Contribution:** 2 fair

**Summary:**

The paper proposes an RL-based model for mixed-size macro placement. The standard cells re placed via gradient-based GPU acceleration and a one-shot conditional generative routing model is devised to perform one-shot routing to the pins within each net. The paper is the first joint placement and routing network without involving any traditional heuristic solver.

**Questions:**

The reviewer feels this paper is only half-baked, not being sufficient for a publication. I would suggest improving several items before a resubmission:

1. Address clearly which part is novel in this paper and which part is just an extension of an existing work. Add the joint placement and routing work in your related work and compare with their work properly.

2. Improve the evaluation section by first, improving the wire length and density results to SoTA baselines like DeepPlace and Graph Placer [2].

3. Ablation on the decomposition of gains and help readers understand what part of the design contributes most to the improved wire length and density. For example, try to show the generative routing actually improves the performance compared to a more general approach using deep RL.

[2] https://arxiv.org/abs/2004.10746

**Limitations:**

Yes.

**Strengths And Weaknesses:**

Strengths:
- Applying generative models for placement and routing is an interesting topic. Even though, generative models have been successfully applied to various domains, however, this work is novel because no generative models have been devised and applied to the pin routing problem before.

Weaknesses:
- The paper makes a false claim that this paper is the first one to address placement and routing end-to-end. But there is a prior work jointly optimizing placement and routing in NeurIPS 2021 [1]. Please correct the statement and add the citation in the related work.

- The paper looks like an extension of an existing work combing placement and routing in an end-to-end fashion. The real novelty is in its generative model devised for the routing problem. This could lead to the bad logical order of the paper, where routing comes first while placement comes last. The EM algorithm is section 3.3 is almost a one sentence summary. If the EM algorithm is also part of the originality of this paper, the authors of this paper should provide more details and formulation for section 3.3. Also, logically, you should consider moving section 3.3 earlier.

- There is a very limited set of comparisons to baselines. Only DeepPlace is compared against the proposed method. Table 1 wire length results look even worse than DeepPlace. Please explain why a worse wire length results would justify publishing the paper?

[1] "On Joint Learning for Solving Placement and Routing in Chip Design", https://papers.nips.cc/paper/2021/file/898aef0932f6aaecda27aba8e9903991-Paper.pdf

---

> ### Author Response · Authors · 2022-08-02
> **Rebuttal to Reviewer KgYJ**
>
> Thank you for the detailed feedback and recognition to our work's novelty and potential influence. Below we respond to your specific comments and we do hope you would re-consider the rating.
>
> **Responses to the main concerns and questions:**
>
> **Q1. *"The paper makes a false claim that this paper is the first one to address placement and routing end-to-end. But there is a prior work jointly optimizing placement and routing in NeurIPS 2021 [1]. Please correct the statement and add the citation in the related work."***
>
> **Re**: Thanks. Please kindly note that what we claimed is that our method is the first **pure** neural networks for placement and routing (see line 58), not claiming the first one to address placement and routing end-to-end as done in [1]. In fact, we carefully compared our work with DeepPlace (i.e., NeurIPS 2021) [1] in footnote 2 (on page 2) to assure our statement (perhaps the footnote is easy to miss out). We also cite this work in the related work section at line 69-70, indicating they propose a joint learning technique for placement and routing.
>
> Thank you for pointing out this potential ambiguity. We will further clarify this statement by emphasizing that we are not the first end-to-end work as done in [1] but (one of) the first pure neural pipelines, and TBH we have not identified any other pure neural P&R pipeline.
>
> **Q2. *"... The real novelty is in its generative model devised for the routing problem. This could lead to the bad logical order of the paper, where routing comes first while placement comes last."***
>
> **Re**: Thanks for your recognition to the novelty of our generative routing model. While we also state in our contribution (see line 54-56) that we also devise a novel mixed-size macro placement learning approach and show its advantage over [1] (see Table 1). In contrast, existing mixed-size placement methods are almost handcrafted heuristic methods instead of learning-based techniques (see [53, 54] in the reference list). In fact, this is ignored in [1]. Our ablation studies also show the effectiveness of our adaptive net routing order learning scheme (see Table 3) which is also new in literature.
>
> **Q3. *"Address clearly which part is novel in this paper and which part is just an extension of an existing work. Add the joint placement and routing work in your related work and compare with their work properly."***
>
> **Re**: As we summarized at line 49-60, the novelty of our work lies in the RL-based model for learning mixed-size placement as described in Section 3.1, the generative model for routing as described in Section 3.2, and finally the neural macro placement and routing pipeline for combining these two parts (including the net order module) in Section 3.3. We hope that novelty can be better delivered by our response and we will also improve the presentation. Thanks for your suggestion.
>
> As answered in Q1, we already cited the previous work DeepPlace (i.e., NeurIPS 2021) [1] in the related work section at lines 69-70. The comparison between our work and DeepPlace [1] is summarized and analyzed in Table 1 and Table 3, and Figure 7 in Appendix.
>
> **Q4. *"... Table 1 wire length results look even worse than DeepPlace. Please explain why a worse wire length results would justify publishing the paper?"***
>
> **Q5. *"Improve the evaluation section by first, improving the wire length and density results to SoTA baselines like DeepPlace and Graph Placer [2]."***
>
> **Re to Q4 & Q5**: We have given an explanation at line 298-302 in the paper that our mixed-size macro placer achieves approximately $4 \times$ reduction over DeepPlace on the overlapping area, with only a 1.3\% increase of total wirelength. These two metrics (i.e., overlap, and wirelength) are mutually contradictory and focusing on wirelength alone is meaningless since we can place macros as close as possible so that the wirelength is 0 while the overlap is extremely large. The advantages of our mixed-size placer are twofold. On the one hand, the severe overlap issue in DeepPlace requires post-processing during detailed placement to resolve collision. Therefore, the final wirelength when routing completes may become much higher than its approximation (i.e., HPWL) in Table 1. On the other hand, our proposed mixed-size placer is able to generate more realistic placement results with higher routability to achieve high-quality routing solutions.

---

> > ### Author Response · Authors · 2022-08-02
> > **Rebuttal to Reviewer KgYJ (Cont.)**
> >
> > As for the evaluation metric, both density and overlapping area are implicit routability models, and density has a positive correlation with the overlapping area. Hence, we adopt HPWL as the proxy of wirelength and overlapping area to evaluate placement results as shown in Table 1. Considering that neither DeepPlace [1] nor Graph Placer [2] considers the size of macros in the formulation while DeepPlace [1] outperforms Graph Placer [2] in the experiments, we believe that DeepPlace [1] is a stronger SoTA baseline.
> >
> >
> > **Q6. *"Ablation on the decomposition of gains and help readers understand what part of the design contributes most to the improved wire length and density. For example, try to show the generative routing actually improves the performance compared to a more general approach using deep RL."***
> >
> > **Re**: As mentioned in our response to Q2 and Q3, our methodology consists of three parts: an RL-based model for learning mixed-size placement, a generative model for routing, and finally the neural macro placement and routing pipeline (including the net order module). Accordingly, we perform ablation studies on each part to investigate which module contributes most to the performance. For our mixed-size macro placer, we show that it achieves approximately $4 \times$ reduction over DeepPlace on the overlapping area (which has a positive correlation with density) by modeling the macro’s shape in state space. In addition, the reduced overlapping area requires less post-processing to resolve the collision, which facilitates the improvement of wirelength in the long term as described at line 298-302 in section 4.2. Similarly, for the generative routing model, we find that the ResNet generator outdoes the cGAN, but the bi-discriminator significantly improves the generator. Moreover, the enhanced loss improves the wirelength at the marginal expense of correctness as described at line 318-320 in section 4.3. Finally, for neural macro placement and routing pipeline, we show there is a significant performance gap in routing congestion without net order learning, and training the placement model with the generative neural router in an end-to-end manner (with EM) further improves the final wirelength as described at line 337-342 in section 4.4.
> >
> > [1] R. Cheng and J. Yan, “On joint learning for solving placement and routing in chip design,” NeurIPS, 2021.
> >
> > [2] A. Mirhoseini, A. Goldie, M. Yazgan, J. W. Jiang, E. Songhori, S. Wang, Y.-J. Lee, E. Johnson, O. Pathak, A. Nazi et al., “A graph placement methodology for fast chip design,” Nature, 2021. (This is the official version of https://arxiv.org/abs/2004.10746.)

---

> > > ### Comment · Reviewer_KgYJ · 2022-08-09
> > > **Thanks for the detailed rebuttal.**
> > >
> > > The response address many of my questions. I am glad to see a growing interest in ML for chip placement and routing and believe it is going to be an important area. I will raise my score by one, yet only by one for a couple of reasons:
> > > 1.  The claimed most novel "mixed-size macro placement" can be still a very incremental contribution, particularly to Neurips community. Compared to the very first chip placement paper, the reviewer feels less thrilled about this paper, even it presents reasonable design flow and evaluation results.
> > >
> > > 2. Logically, this paper is not straightforward to read, as it flips the order of placement and routing. The introduction to GAN does not seem to be super relevant. And the results does not show immediate gains.
> > >
> > > Thank the authors timely reply. Good luck.

---

> > > > ### Author Response · Authors · 2022-08-09
> > > > **Reply to Reviewer KgYJ**
> > > >
> > > > Thanks for your valuable and constructive comments. We also agree with your current two points and we have tried to polish the logic of the paper in the newly submitted version. We assure that in our new version routing is always discussed after placement and finally joint placement and routing pipeline is discussed. Thank you again for your feedback which greatly improves our work.
> > > >
> > > > Back to your first concern in terms of significant novelty, in our humble opinion it is not surprise as this is an application-driven paper. But we would emphasize the progress we made in the generative routing part, as you can see from Table 9 in appendix, our new method achieves competitive wirelength while the speed can outperform a few traditional routers. We believe with the fast growth of GPU resource, its advantage can be further tapped. Hence we believe our work can have more impact in this emerging area. From the practical perspective, mixed-size placement is critical to make the machime learning placer really work, and this paper is an early published effort. We have uploaded our code in supplementary material (note Google does not support mixed-size placement in their source code and is only compatible for Ariane RISC-V dataset while while our code interface follows the widely used LEF/DEF format which can be readily applied to most public dataset) to promote the progress in this community. We will further clean our code in final version.

---

> ### Author Response · Authors · 2022-08-06
> **Waiting for Reply**
>
> Dear reviewer KgYJ:
>
> We thank you for the precious review time and valuable comments. We have provided corresponding responses and results, which we believe have covered your concerns. We hope to further discuss with you whether or not your concerns have been addressed. Please let us know if you still have any unclear parts of our work.
>
> Best,
>
> Authors of Paper5574

---

### Author Response · Authors · 2022-08-08
**Additional Experimental Results**

We finish the remaining two chips in the ISPD-2005 benchmark suites and in table below we show results on Joint Macro Placement and Routing pipeline, which is extension to Table 3.

|                                                     | adaptec4           |                    | bigblue3           |                    |
|-----------------------------------------------------|--------------------|--------------------|--------------------|--------------------|
| variants of our PRNet                               | **WL$\downarrow$** | **RC$\downarrow$** | **WL$\downarrow$** | **RC$\downarrow$** |
| RL-based Placer (i.e. DeepPlace)                |          47933        |  128.257  | 42550               | 67.124             |
| RL-based Placer + GR                                |         47406         |       121.435           | 41617               | 70.187             |
| RL-based Placer + GR + NOL (full version of PRNet ) |         **46964**         | **65.796**   | **40509**               | **64.964**                 |

Since the discussion phase is approaching the end, we are looking forward to receiving comments from reviewer KgYJ.

---

### Meta-Review · Area_Chair_5ota · 2022-08-27

**Recommendation:** Accept
**Confidence:** Less certain

**Metareview:**

This paper proposes a neural network pipeline for chip design that utilizes reinforcement learning for mixed-size macro placement and a conditional generative routing network that can perform routing in one shot. While the reviewers had some reservations regarding baselines, sensitivity to hyperparameters, and the resulting wirelength compared to DeepPlace, these have largely been addressed in the rebuttals and discussion. It was generally felt that this approach is quite novel and effective. It would be good to put the extra results in the final draft.

**Award:**

No

---

### Decision · Program_Chairs · 2022-09-14

Accept